# Boost Protein Language Model with Injected Structure Information through Parameter Efficient Fine-tuning

## Abstract

At the intersection of computer vision and computational biology, large-scale Protein Language Models (PLMs), particularly the ESM series, have made significant advances in understanding protein structures and functions. However, these models are mainly pre-trained on pure residue sequence, often lack explicit incorporation of structural information, highlighting an opportunity for enhancement. In this paper, we design a parameter-efficient fine-tuning method, SI-Tuning, that injects structural information into PLMs while preserving the original model parameters frozen and optimizing a minimal task-specific vector for input embedding and attention map. This vector, extracted from structural features like dihedral angles and distance maps, introduces a structural bias that enhances the model's performance in downstream tasks. Extensive experiments show that our parameter-efficient fine-tuned ESM-2 650M model outperforms SaProt, a large-scale model pre-trained with protein structural data, in various downstream tasks with a reduction of 40.3% GPU memory and 39.8% time consumption.

## 1 Introduction

Proteins, the molecular backbones of life, orchestrate an array of vital functions, facilitating complex cellular activities and biological mechanisms. In the realm of computational biology, the conceptual merger with Natural Language Processing (NLP) techniques has catalyzed the emergence of Protein Language Models (PLMs), as elucidated by Rao et al. (2019). These innovative models adeptly navigate the vast landscapes of protein sequences through self-supervised learning, exhibiting a remarkable capability to unravel the intricate patterns of residue interactions reflective of co-evolutionary processes, a domain extensively explored by Anishchenko et al. (2017) and Rao et al. (2020). In this cutting-edge arena, models such as UniRep (Alley et al., 2019), ProtTrans (Elnaggar et al., 2021), ESM (Rives et al., 2021; Meier et al., 2021; Rao et al., 2021; Lin et al., 2022), and Evoformer (Hu et al., 2022; Jumper et al., 2021) stand out, having been rigorously pre-trained on comprehensive datasets. Their exceptional generalization skills have been pivotal in advancing research across a spectrum of downstream tasks central to deciphering protein structures and functions. These encompass structure prediction, functional annotation, and protein-protein interactions, alongside the innovative field of protein design, as evidenced by notable works (Jumper et al., 2021; Gligorijević et al., 2021; Corso et al., 2022; Hsu et al., 2022). Leveraging Parameter-Efficient Fine-Tuning (PEFT) methods could significantly enhance downstream tasks on these PLMs performance with only a minimal increase in trainable parameters.

In the context of VLMs and LLMs, PEFT techniques like LoRA (Hu et al., 2021), Prefix-tuning (Li & Liang, 2021), and Prompt tuning (Lester et al., 2021) are fundamentally akin to feature projection processes. Different from language or vision embedding sequences, protein sequences, characterized by their unique primary structures, lack the spatial detail inherent in their three-dimensional conformations, which are pivotal for functional diversity. Notably, the prominent protein language model ESM is predicated solely on sequence information, treating amino acid residues as tokens without integrating three-dimensional structural data. The recent innovation, SaProt (Su et al., 2023), introduced a Structural Awareness (SA) lexicon, merging primary and tertiary structural information into SA tokens, thereby suggesting an advanced modality for protein representation. Then SaProt utilizes the ESM-2 framework, a universal PLM was developed on this premise. Despite

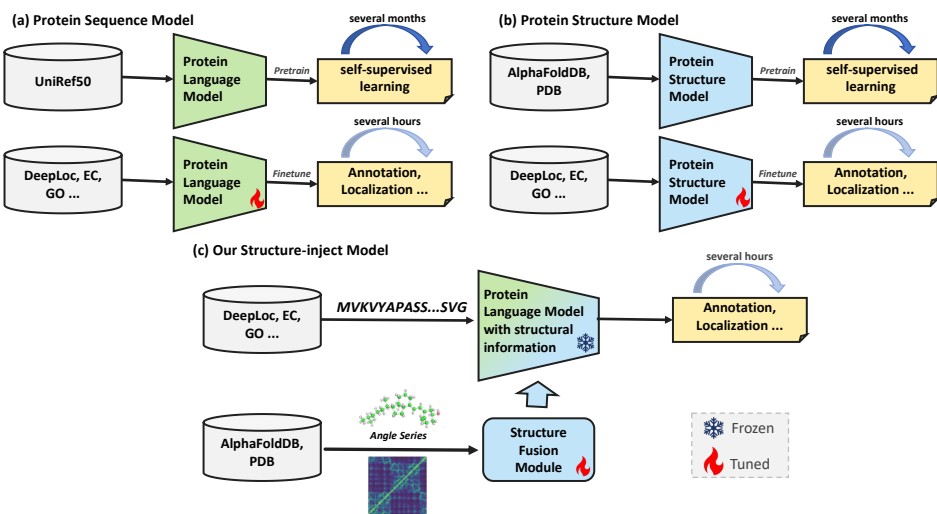

Figure 1: Comparison of traditional protein modeling pipelines and our proposed pipeline. (a) Protein language models capture structural, functional, and evolutionary information by processing amino acid sequences. These models require self-supervised training on extensive protein sequence datasets for weeks or even months, followed by full-parameter fine-tuning on downstream tasks. (b) Protein structure models build upon language models by incorporating structural information. They process not only one-dimensional sequence data but also consider the three-dimensional spatial arrangements of amino acid residues. Similar to protein language models, they demand months of training time and substantial computational resources. (c) Our proposed SI-Tuning method injects structural information into protein language models without the need for pre-training by leveraging advanced Parameter-Efficient Fine-Tuning (PEFT) techniques. Compared to the traditional pre-training and full-parameter fine-tuning workflow, our approach significantly reduces computational time and resource requirements while maintaining strong performance on downstream tasks.

SaProt's attempt to incorporate 2D structural embeddings, it essentially implicitly learns structural information, with training being resource and data intensive.

To circumvent these limitations, we propose the Structure information Injecting Tuning (SI-Tuning), to fully leverage structural insights in the efficient fine-tuning stage to refine PLMs for downstream tasks. Capitalizing on the precise structural predictions provided by AlphaFold2, our SI-Tuning attempts to embed this comprehensive structural information produced by AlphaFold2 within the PEFT paradigm, as shown in Fig. 1. To be specific, we design two kinds of injection strategy, embedding injection and attention map injection, for different forms of structural information. For individual level structural information (angle series), we generate an angular embedding via self-gate module and then fuse with the original sequence embedding. While for pairwise structural information (distance maps), we transform the distance maps by radial basis functions and then inject this pairwise correlation into attention matrix of each attention block. Based on these two injection strategies, we simply employ the commonly used LoRA (Hu et al., 2021) as our PEFT adaption learner to efficiently tune the PLMs. This allows the large models to acquire richer structural information, leading to a better performance.

In summary, our contributions are three-folds:

- We introduced a PEFT approach that incorporates structural information into PLM, called Structural information Injecting Tuning (SI-Tuning), which could effectively address the explicit lack of structural comprehension within PLMs and could efficiently fine-tune PLM for downstream tasks.

- In our SI-Tuning, we incorporate sequential and pairwise structural information, dihedral angle series and distance maps, into PLM via embedding injection and attention map in-

jection respectively. During training, we efficiently fine-tune the PLM with LoRA to help the PLM explicitly learn the structural information for downstream tasks.

- Extensive experiments confirmed the effectiveness of our approach. Compared to full tuning ESM-2, our SI-Tuning only needs less than 2% tunable parameters for downstream tasks while keeping ESM-2 backbone frozen. Specifically, our SI-Tuning achieves an improvement of 4.49% and 1.99% on Metal Ion Binding dataset and DeepLoc binary classification task compared with full model tuning.

## 2 RELATED WORK

### 2.1 PROTEIN LARGE MODELS

#### 2.1.1 SEQUENCE-BASED REPRESENTATION

Sequence-based protein representation learning conceptualizes protein sequences within the framework of natural language processing. Recent endeavors have drawn inspiration from large pre-trained language models, embarking on a self-supervised learning journey across billions of protein sequences to decipher evolutionary information. These sequence-based protein large models (PLMs) conform to the paradigm established by BERT (Devlin et al., 2018) (Bidirectional Encoder Representations from Transformers), regarding individual protein sequences as sentences and amino acid residues as words. By randomly masking a proportion of residues and deploying multiple layers of self-attention and feedforward networks, these models engage in masked language modeling to predict obscured residues, thereby capturing the interdependencies between residues.

Proteins represented by sequence-based PLMs exhibit commendable performance across a variety of downstream tasks. Harnessing a wealth of unlabeled data, the ESM family—comprising contributions from Rives et al. (2021); Lin et al. (2023) —has been adept at capturing the evolutionary relationships among proteins, identifying conserved patterns across different species that are essential for biological functionality, and applying these insights to tasks such as protein classification, structure prediction, and interaction prediction. Our paper utilizes the advanced ESM-2 (Lin et al., 2023) model as the backbone. Furthermore, the ESM-1v, developed by (Meier et al., 2021), is a variant within the Evolutionary Scale Modeling (ESM) family of protein language models. It's designed to predict the impacts of mutations on protein functions. Research by Rao et al. (2020) has demonstrated that Transformer attention maps are capable of learning contacts from unsupervised language modeling tasks. Elnaggar et al. (2021) trained various Transformer-based models on extensive protein sequence data from the UniRef and BFD databases, encompassing up to 393 billion amino acids. Rao et al. (2021); Biswas et al. (2021); Meier et al. (2021) enhanced the capabilities of Protein Language Models (PLMs) by leveraging training on Multiple Sequence Alignment (MSA) data, which enriches models with evolutionary context and functional insights from a diverse array of protein sequences.

#### 2.1.2 STRUCTURE-BASED REPRESENTATION

The functional attributes of proteins are intrinsically determined by their structures, making structure-based approaches fundamentally superior for learning rich protein representations compared to sequence-based methods. The advent of AlphaFold2 (Jumper et al., 2021) has revolutionized protein structure prediction, enabling access to extensive structural data. Proteins are often modeled as graphs with amino acids as nodes and chemical bonds as edges, employing graph neural networks for representation learning through techniques such as contrastive learning and denoising score matching (Hermosilla & Ropinski, 2022; Chen et al., 2023; Guo et al., 2022). GearNet (Zhang et al., 2022) introduces a novel Step-wise Dual Learning approach for addressing the weakly supervised domain adaptation (WSDA) problem, refining the edge representation of proteins. Another noteworthy development is SaProt (Su et al., 2023), which has shown superior capability in capturing the geometric configurations of proteins by integrating a structure-aware vocabulary from 3D protein structures, albeit at the cost of significant computational resources and time.

## 2.2 PARAMETER-EFFICIENT FINE-TUNING

Large-scale pre-trained models have achieved significant success in both natural language processing (Devlin et al., 2018; Qiu et al., 2020) and computer vision fields (Dosovitskiy et al., 2020), reaching state-of-the-art performance across various tasks. However, the conventional approach of fine-tuning all parameters of pre-trained models becomes less feasible with the increase in model size and task quantity, as retraining all model parameters for complete fine-tuning is impractical. To mitigate this issue, various parameter-efficient transfer learning methods have emerged, with the PEFT(Parameter-Efficient Fine-Tuning) method fine-tuning only a small number of parameters to achieve robust performance. Several popular lightweight PEFT schemes are now available.

Adapter tuning (Houlsby et al., 2019) introduced small neural modules called adapters between layers of the pre-trained model. Each adapter consists of a down-projection to a lower-dimensional space, followed by a non-linear activation function and an up-projection back to the original dimension, with only the parameters within these adapter modules being updated during fine-tuning. Prompt tuning, introduced by Lester et al. (2021), is a parameter-efficient method that fine-tunes pre-trained models by attaching trainable vectors (referred to as prompts) to the input embeddings. Unlike methods that modify the model architecture or require training a large number of parameters, prompt tuning adjusts only these additional vectors while keeping the original model parameters frozen. Inspired by the success of prompt-based learning, Li & Liang (2021) proposed prefix tuning, which involves adding a set of tunable prefix vectors to the keys and values of the multi-head attention layer in each transformer block. These prefix vectors are optimized during fine-tuning, while the original pre-trained parameters remain unchanged. Lora, proposed by Hu et al. (2021), modifies the attention and feed-forward layers of the transformer by learning a low-rank matrix approximating the updates to the original weight matrices. The original weights are kept frozen, and only the low-rank matrices are updated during fine-tuning.

Currently, the application of PEFT techniques to large protein models is relatively scarce. PEFT-SP (Zeng et al., 2023) utilizes methods like LoRA and Adapter Tuning in the ESM-2 model to better leverage evolutionary knowledge from protein sequences, leading to more accurate predictions of signal peptides in proteins. Another approach (Sledzieski et al.; 2023) employs LoRA for training models to predict protein-protein interactions and homooligomer symmetry, demonstrating performance that either matches or exceeds traditional fine-tuning methods while significantly reducing memory and parameter usage. Current PEFT applications in protein large models are tailored to specific tasks. Our research aims to develop a more generalized PEFT paradigm, incorporating protein structural information into large protein language models to enhance performance across a wide range of downstream tasks.

## 3 METHOD

As shown in 2, our structural information injecting tuning (SI-tuning) provides a paradigm of Parameter-Efficient Fine-Tuning (PEFT) for large protein language models.

Based on pre-trained PLM, we first obtain the sequence information, individual level structural information (angle series) and pairwise structural information (distance map) of the input protein. Then, the angular embedding generated by angle series is injected into the original sequence embedding. This results in a protein embedding enriched with geometric information. This enhanced embedding is then fed into the PLM encoder comprising $N$ blocks. The pairwise relationship of amino acid (protein contact map), processed through radial basis functions (RBF), is injected into each attention block, interacting with the attention matrix to enhance the model's structural awareness. Finally, the feature vectors produced by the encoder are passed through an output head tailored for various downstream tasks.

For a fair comparison, we employ the ESM classification head which consists of a dense layer, followed by dropout for regularization, a nonlinear activation function (tanh), and a final linear projection layer that maps the processed features to the desired number of output labels. During training, we employ LoRA as the base parameter-efficient fine-tuning method to better integrating structural information into pre-trained large protein language models.

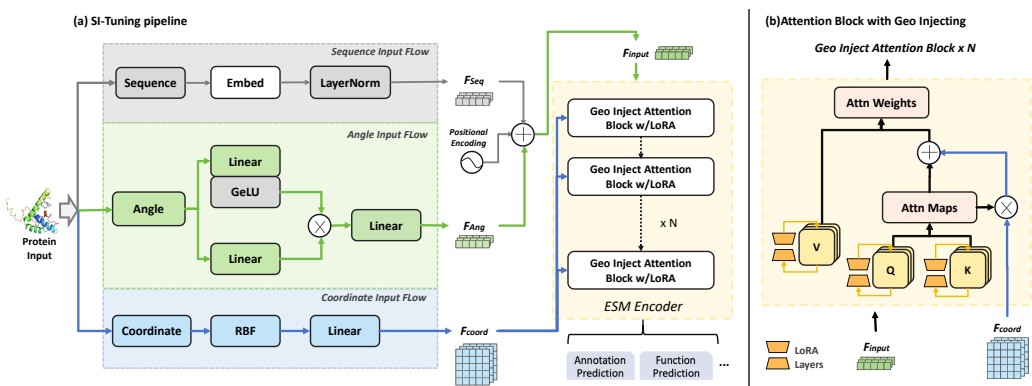

Figure 2: SI-tuning Architecture. (a) Our training pipeline enhances the protein language model with two additional structural information modalities beyond the protein sequence features $F_{\text{seq}}$. The angle series of the protein, effectively integrated via self-gating, yields serialized structural features $F_{\text{angle}}$. These features, combined with $F_{\text{seq}}$, form $F_{\text{input}}$. The coordinates of the protein are processed through RBF to obtain pairwise geometric features $F_{\text{corr}}$, which are then input into each attention block. (b) $F_{\text{corr}}$ acts as a bias to assist in updating the attention weights, resulting in a structure-aware protein representation.

## 3.1 SEQUENTIAL STRUCTURAL INFORMATION ENHANCED EMBEDDING

In stereo-chemistry, the sequence of dihedral angles and bond lengths is commonly used to represent the polymers backbones, especially proteins, in internal coordinates. This individual level structural information contains sequential features (angle series) for each amino acid residues in protein structure.

### 3.1.1 SEQUENTIAL STRUCTURAL INFORMATION FORMATION

The sequential structural information of proteins can be characterized by several critical angles that define their conformation:

1. **Psi ($\psi$) and Phi ($\phi$) Angles:** These are the dihedral angles along the protein backbone, which dictate the backbone's configuration and influence whether the structure forms alpha helices or beta sheets.

2. **Chi ($\chi$) Angles:** These angles represent the dihedral angles of protein side chains, with each residue having up to four $\chi$ angles. These angles are crucial as they determine the spatial arrangement of protein side chains, impacting the protein's function and structure.

3. **Theta ($\theta$) and Tau ($\tau$) Angles:** These bond angles in the protein backbone describe the geometric shape of the backbone.

4. **pLDDT scores($p$):** This is a confidence score that estimates the reliability of each predicted residue in a protein structure.

We leverage the angular structural information above as the first kind of structural information for each amino acid apart from residue type, $\alpha = \{\psi, \phi, \chi_1, \chi_2, \chi_3, \chi_4, \theta, \tau\} \in \mathbb{R}^{N \times 8}$, where $N$ is the sequence length. Considering the cyclic nature of these angles, we transform these angles with trigonometric function to form the raw angular structural information $x_{angle} \in \mathbb{R}^{N \times 16}$:

$$x_{angle} = [\sin(\psi), \cos(\psi), ..., \sin(\tau), \cos(\tau)].$$

This transformation could ensure the periodic property of angle which is suitable for the subsequent embedding processing. For scenarios where predicted structures are utilized, we further integrate the confidence score pLDDT scores $p$ into the sequential structural information. The pLDDT scores

indicate the confidence levels for each residue's predicted structure, enabling us to appropriately weight the angular information. This integration allows the pLDDT scores to complement the angular structural information, thereby enhancing the model's capacity to incorporate confidence levels into the embedding process:

$$x_{\text{angle}} = [\sin(\psi), \cos(\psi), ..., \sin(\tau), \cos(\tau), p] \in \mathbb{R}^{N \times 17}.$$

### 3.1.2 Angular Embedding Generation and Enhancement

To effectively incorporate structural information into the embeddings of protein sequences, we employ a self-gating mechanism, which is a straightforward yet powerful method for feature projection and filtering. This mechanism processes the raw angular structural information, $X_{angle}$, by applying a series of linear projections followed by nonlinear activations. This process can be formulated as follow:

$$F_{angle} = f(X_{angle}) \odot \sigma(g(X_{angle})),$$

where $f$ and $g$ represent linear transformations, while $\sigma$ denotes the activation function. Here, we utilize the Gaussian Error Linear Unit (GELU), which can be seen of as a smoother variant of ReLU and is commonly used in the most advanced Transformers.

Finally, the resultant angle embedding $F_{angle}$ is fused with the original sequence embedding $F_{seq}$ via weighted sum resulting in an enhanced input embedding:

$$F_{input} = F_{seq} * (1 - \gamma) + F_{angle} * \gamma,$$

where $\gamma$ is a learnable weight that balance the influence of $F_{angle}$ with the original sequence embedding $F_{seq}$.

### 3.2 Enhanced Geometric Encoding with Radial Basis Functions

In the study of molecular representations, topological information such as chemical bonds and interatomic interactions is crucial for understanding molecular structures and properties. Therefore, beyond sequence representations, e.g., amino acid residues and dihedral angles, geometric information is also a vital aspect for representation. In this paper we leverage the distance matrix as the geometric information, the distance map represents the distance between all possible amino acid residue pairs, which is much more concise than 3D atomic coordinates and is invariant to rotations and translations. Inspired by Geoformer (Wang et al., 2024), we design the Geo-Inject Attention to inject the geometric information into pre-trained PLMs, which enables PLMs to perceive geometric information.

#### 3.2.1 Inter-residue Representing

First we utilize Radial Basis Functions (RBFs) to transform pairwise distances between atoms into a format suitable for neural processing. For each pair of residues $i$ and $j$ from a protein of length $L$, their Euclidean distance $d_{ij}$ is transformed using an RBF:

$$F_{coord,ij} = \phi(d_{ij}) \cdot \exp\left(-\beta_{ij} \left(\exp\left(-d_{ij}\right) - \mu_{ij}\right)^2\right),$$

where $d_{ij}$ represents the distance between atoms $i$ and $j$, $\beta_{ij}$ and $\mu_{ij}$ are learnable parameters that specify the center and width of $F_{coord}$, and $\phi(\cdot)$ is a smooth cosine cutoff function.

#### 3.2.2 Geo-Inject Attention

The enhanced pairwise encodings, $F_{coord}$, are integrated into the attention matrix in self-attention block of PLM as a prior knowledge of pairwise relationship. During the training process, we first project the original query and key to $d_{inter}$ and generate the auxiliary attention map $A_{aux} \in \mathbb{R}^{bh \times L \times L \times d_{inter}}$, then the pairwise encodings is transformed through a linear layer followed by a

SiLU activation and injected into the original attention map of self-attention:

$$A_{aux} = \text{Unsqueeze}(f_q(Q)) \cdot \text{Unsqueeze}(f_k(K)),$$

$$\text{Geo}(A_{aux}, F_{coord}) = \sum_{d_{inter}} A_{aux} \cdot \text{SiLU}(f_{coord}(F_{coord})),$$

$$\text{Attention}(Q, K, V) = \text{softmax}\left(\frac{QK^\top}{\sqrt{d_k}} + \text{Geo}(A_{aux}, F_{coord})\right) V,$$

where $Q$, $K$, and $V$ are query, key, and value respectively and $d_k$ represents the dimension of $Q$, $K$ and $V$, while $f_*$ denotes the linear transformation for auxiliary attention map and $F_{\text{coord}}$, and $\cdot$ represents hadamard product. Specifically, the auxiliary attention map $A_{aux}$ generated by broadcast hadamard product can be viewed as the expanded form of attention matrix multiplication:

$$Q_{aux} = \text{Unsqueeze}(f_q(Q)) \in \mathbb{R}^{bh \times L \times 1 \times d_{inter}},$$

$$K_{aux} = \text{Unsqueeze}(f_k(K)) \in \mathbb{R}^{bh \times 1 \times L \times d_{inter}},$$

$$f_q(Q) \times f_k(K)^\top \Leftrightarrow \sum_{d_{inter}} A_{aux} = \sum_{d_{inter}} Q_{aux} \cdot K_{aux}$$

We introduce pairwise information into this expanded attention map by incorporating a structure-specific relationship map. This enhanced expanded attention map is then reduced along the $d_{inter}$ dimension, resulting in a supplementation for original attention matrix which is comprised of correlation of each amino acid pair.

## 3.3 Low-Rank Adaptation for PLM

To better integrating structural embeddings into pre-trained PLM, we leverage a commonly used PEFT method Low-Rank adaptation (LoRA) (Hu et al., 2021) to efficiently adapting PLM to downstream tasks with our structural embeddings ($F_{angle}$ and $F_{corrd}$). It is worth noting that our structural embeddings can be incorporate with various PEFT methods for foundation model. Here we choose LoRA since it do not introduce additional latency during inference for computational efficiency.

To be specific, each projection matrix $W_*$ for the query ($Q$), key ($K$) and value ($V$) in the attention block is augmented with a corresponding low-rank update matrix $\Delta W = BA$. Here, $B \in \mathbb{R}^{d_h \times r}$ and $A \in \mathbb{R}^{r \times d_k}$, where $d_h$ is the dimension of input $X$ and $r$ is the rank which is much less than $d_h$ and $d_k$. The transformation can be formulated as follow:

$$\begin{pmatrix} Q \\ K \\ V \end{pmatrix} = \begin{pmatrix} W_q \\ W_k \\ W_v \end{pmatrix} X + \frac{\alpha}{r} \begin{pmatrix} B_q A_q \\ B_k A_k \\ B_v A_v \end{pmatrix} X,$$

where $\alpha$ is a scaling constant that adjusts the contribution of the low-rank updates relative to the original weight matrices. Following Hu et al. (2021), each $A$ is initialized with a random Gaussian distribution and $B$ is set to zero. This setting ensures that $\Delta W = BA$ starts from zero, preserving the original learned parameters of the ESM-2 model at the beginning of training. During training, the pre-trained weights $W_*$ are frozen.

## 4 Experiments

### 4.1 Datasets

In this study, we concentrate on the adaptation of pre-trained PLMs to supervised downstream tasks. The downstream task configuration follows the setting used in SaProt (Su et al., 2023), the details of datasets are shown in Tab. 1.

**Protein Function Prediction:** This aspect leverages the "human cell" segment from the FLIP dataset's thermostability task (Elnaggar et al., 2021), aimed at ascertaining the thermostability metrics of proteins. Moreover, the metal ion binding task Hu et al. (2022) is employed to ascertain the presence of metal ion binding sites within protein structures.

Table 1: Details of downstream task datasets.

| Dataset | Category | Evaluation Metric | Train | Valid | Test |
|---|---|---|---|---|---|
| Thermostability | - | Spearman's $\rho$ | 5056 | 639 | 1336 |
| Metal Ion Binding | - | ACC (%) | 4247 | 1066 | 1083 |
| EC | - | Fmax | 13089 | 1465 | 1604 |
| GO | BP / MF / CC | Fmax | 26224 | 2904 | 3350 |
| DeepLoc | Binary | ACC (%) | 5477 | 1336 | 1731 |
| | Subcellular | ACC (%) | 8747 | 2191 | 2747 |

**Protein Annotation Prediction:** DeepFRI benchmark (Gligorijević et al., 2021) predicts protein annotations laden with multifarious functional labels, we choose Enzyme Commission (EC) numbers and Gene Ontology (GO) terms. The GO benchmark amalgamates predictions across Molecular Function (MF), Biological Process (BP), and Cellular Component (CC) domains.

**Protein Localization Prediction:** DeepLoc dataset (Almagro Armenteros et al., 2017) endeavors to predict the subcellular localization of proteins. DeepLoc introduces a bifurcated approach to subcellular localization, with one branch categorizing into 10 distinct localizations and another adopting a binary classification scheme.

## 4.2 IMPLEMENTATION DETAILS

We implement our SI-Tuning based on the open source repository of ESM [1] released by Facebook. All the experiments are conducted with $1\times$ Nvidia 4090 for 35M ESM-2 and 650M ESM-2. For the Thermostability, Metal Ion Binding, and DeepLoc tasks, we set the learning rates and batch size for our SI-Tuning to 0.0000625 and 1, respectively. For EC tasks, MIBind. tasks and GO tasks, the learning rates and batch size of our SI-Tuning approach are set to 0.0002 and 1 respectively. We found that low learning rates can lead to bad results on, this may be caused by the noise of real structures. We utilize the AdamW optimizer with a weight decay of $10^{-8}$ and a StepLR scheduler. We train our SI-Tuning for 100 epochs in total. During training, the encoder part of the pre-trained PLM is frozen, we only tune our structural encoding modules and LoRA modules.

Table 2: Evaluation results of SI-Tuning under ESM-2 650M. *Therm.* and *MIBind.* denote Thermostability and Metal Ion Binding datasets, respectively. *Bin.* and *Sub.* represent Binary and Subcellular, respectively. In contrast to our SI-Tuning, other comparison methods require full-parameter fine-tuning, consuming several times more time and computational resources. The best score is marked by red, while the second best score is marked by blue.

| Model | Therm. | MIBind. | EC | GO | | | DeepLoc | |
|---|---|---|---|---|---|---|---|---|
| | | | | MF | BP | CC | Bin. | Sub. |
| | $\rho$ | ACC | Fmax | Fmax | Fmax | Fmax | ACC | ACC |
| ESM-2 | 0.680 | 71.56 | 0.868 | 0.670 | 0.473 | 0.470 | 91.96 | 82.09 |
| ESM-1b | 0.708 | 73.57 | 0.864 | 0.656 | 0.451 | 0.466 | 92.83 | 80.33 |
| MIF-ST | 0.694 | 75.08 | 0.807 | 0.633 | 0.375 | 0.322 | 91.76 | 78.96 |
| GearNet | 0.571 | 71.26 | 0.874 | 0.644 | 0.481 | 0.476 | 89.18 | 69.45 |
| ESM-GearNet | 0.651 | 74.11 | 0.887 | 0.676 | 0.516 | 0.507 | 92.94 | 82.30 |
| SaProt | 0.724 | 75.75 | 0.882 | 0.682 | 0.486 | 0.479 | 93.55 | 85.57 |
| SI-Tuning | 0.703 | 76.05 | 0.888 | 0.671 | 0.478 | 0.496 | 93.95 | 84.86 |

## 4.3 MAIN RESULTS

The overall results are shown in Tab. 2. Our SI-Tuning achieves the best results in Metal Ion Binding, EC, and DeepLoc binary classification task, while obtaining the second-best performance in GO-CC and DeepLoc subcellular classification task. Compared to the full-parameter fine-tuned

---

[1]https://github.com/facebookresearch/esm

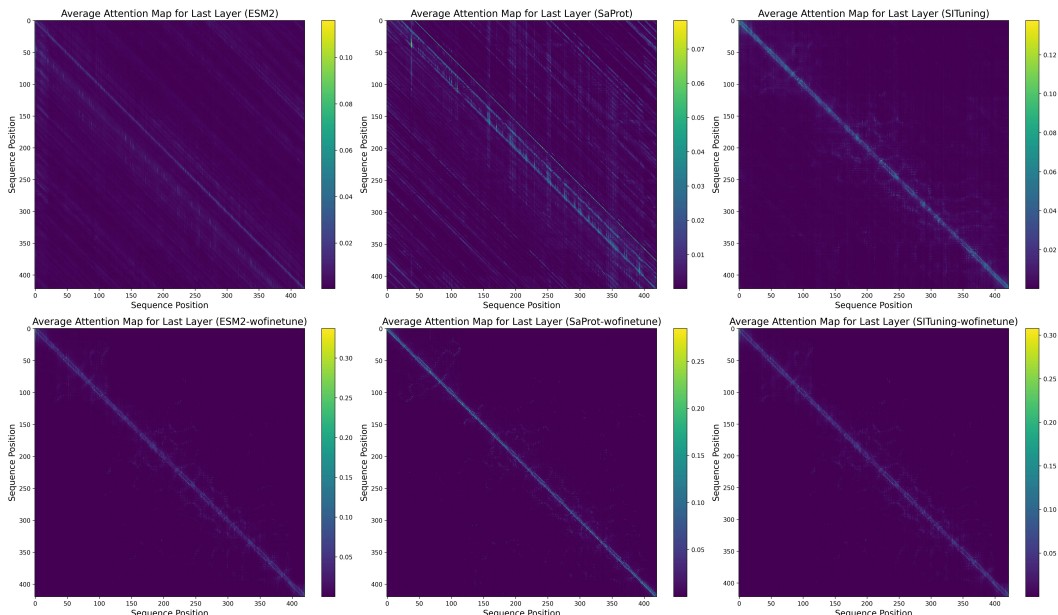

Figure 3: Visualization of attention map of full-parameter tuned ESM-2, SaProt, and our parameter-efficient SI-Tuning.

Table 3: Results of SI-Tuning w/ and w/o pLDDT scores on ESM-2 models.

| Method | pLDDT | Thermostability | DeepLoc-Bin. | DeepLoc-Sub. |
|---|---|---|---|---|
| SI-Tuning 35M | ✗ | 0.687 | 92.03 | 77.47 |
| SI-Tuning 35M | ✓ | 0.689 (+0.002) | 92.79 (+0.76) | 79.65 (+2.18) |
| SI-Tuning 650M | ✗ | 0.689 | 93.42 | 84.49 |
| SI-Tuning 650M | ✓ | 0.703 (+0.014) | 93.95 (+0.53) | 84.86 (+0.37) |

ESM-2, our SI-Tuning improves regression task performance by 0.001 to 0.026 and achieves an improvement of 1.99% to 4.49% in classification tasks with less than 2% tunable parameters. In comparison to SaProt, which is pre-trained with large-scale structure-aware vocabulary, our SI-Tuning achieves comparable or better performance in three classification tasks. While for regression tasks, our SI-Tuning surpasses SaProt by 0.006 and 0.019 in EC and GO-CC, while is slightly weaker in Thermostability, GO-MF and GO-BP.

To provide a clearer view of the advantages of our method, we present the attention map of the last block of full tuned ESM-2, SaProt, and our SI-Tuning in Fig. 3. The visualization reveals that ESM-2 and SaProt without fine-tuning on the downstream datasets, exhibit a potential capacity for structural awareness, as indicated by the presence of certain contact information. However, after full-parameter fine-tuning, the structural contact information appears to be destroyed on ESM-2 and SaProt. We conjecture that this degradation may be due to the limited data in downstream tasks, causing PLM to overfit to specific residue sequence information. In contrast, by freezing the PLM backbone during fine-tuning and incorporating structural information, our SI-Tuning further enhances the model's structural perceptual ability, resulting in a more pronounced contact pattern in the attention map. We provide more visualization results in the appendix for further insight.

## 4.4 ABLATION STUDIES

### 4.4.1 IMPACT OF PLDDT SCORES

Considering Thermostability and DeepLoc, lack native structural information, we conduct several experiments to verify the impact of pLDDT scores on the performance of SI-Tuning. The results are shown in Tab. 3. It can be observed that after appending pLDDT scores, SI-Tuning improves

Table 4: Performance of different LoRA Ranks on Thermostability and DeepLoc datasets.

| LoRA Rank | Turnable Param. | Thermostability | DeepLoc-Bin. | DeepLoc-Sub. |
|-----------|-----------------|-----------------|--------------|--------------|
| 4 | 3.09M (0.47%) | 0.713 | 90.07 | 84.42 |
| 8 | 6.16M (0.93%) | 0.718 | 91.40 | 83.07 |
| 16 | 12.31M (1.84%) | 0.703 | 93.95 | 84.86 |
| 32 | 24.59M (3.61%) | 0.695 | 92.97 | 82.05 |

Table 5: Results of our method with different structure information.

| Method | Angle | Coord. | Therm. | MIBind. | DeepLoc-Bin. | DeepLoc-Sub. |
|-----------|-------|--------|--------|---------|--------------|--------------|
| Full tuning | ✗ | ✗ | 0.680 | 71.56 | 91.96 | 82.09 |
| SI-Tuning | ✓ | ✗ | 0.701 | 74.55 | 93.13 | 84.76 |
| SI-Tuning | ✗ | ✓ | 0.691 | 72.90 | 93.13 | 83.77 |
| SI-Tuning | ✓ | ✓ | 0.703 | 76.05 | 93.95 | 85.40 |

the performance on Thermostability by 0.002 and 0.014 for the 35M and 650M ESM-2 models, respectively. While on DeepLoc, SI-Tuning also achieves improvement with pLDDT scores, with increases of 0.76 and 0.53 for binary classification task and 0.57 and 0.46 for subcellular classification task on the 35M and 650M models, respectively. In the appendix, we present the accuracy improvements of our SI-Tuning across different pLDDT score intervals for comparison.

### 4.4.2 IMPACT OF LoRA RANK

The LoRA Rank is a critical hyperparameter in LoRA fine-tuning, determining the number of parameters that can participate in learning. We conduct experiments on the ESM-2 650M model for Thermostability and DeepLoc with ranks of $\{4, 8, 16, 32\}$. The results are presented in Tab. 4. It can be observed that the optimal rank varies across different datasets. The overall performance peaks at $rank = 8$ for Thermostability and at $rank = 16$ for DeepLoc. In summary, our SI-Tuning can efficiently tune the PLM for downstream tasks with less than 2% tunable parameters.

### 4.4.3 IMPACT OF DIFFERENT STRUCTURAL INFORMATION

To verify the impact of different structural information, we conduct several experiments on ESM-2 650M with SI-Tuning. The results are shown in Tab. 5. Compared to full-parameter fine-tuning, our SI-Tuning with structural information consistently improved performance across all datasets. Specifically, SI-Tuning enhances the ESM-2 by 0.021 and 0.011 through the incorporation of angular and distance map information on Thermostability, respectively. When combining both types of structural information, SI-Tuning further achieves an improvement of 0.023. The results indicate that angular information provides a slightly better improvement to the PLM compared to distance map information. Across all datasets, the combination of angular information and distance map information yields the greatest improvement for the PLM with our SI-Tuning.

## 5 CONCLUSION

In this paper, we propose Structural information injection tuning (SI-Tuning), a parameter-efficient fine-tuning framework, for PLM fine-tuning. Considering the lack of structural comprehension of current PLMs, we introduce two injection strategies to integrate different kinds of structural information into pre-trained PLM. To be specific, for individual level structural information, i.e. angle series, we fuse the angular embeddings into the original sequence embeddings. While for pairwise relationship of amino acid residues, i.e. distance maps, we first transform these maps with radial basis functions and inject this correlation into the attention matrix of each attention blocks. Combining these injection strategies, we efficiently fine-tune the PLM with LoRA modules for downstream tasks. Extensive experiments on five downstream datasets confirmed the effectiveness of our approach.

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
