# A  MORE VISUALIZATION RESULTS

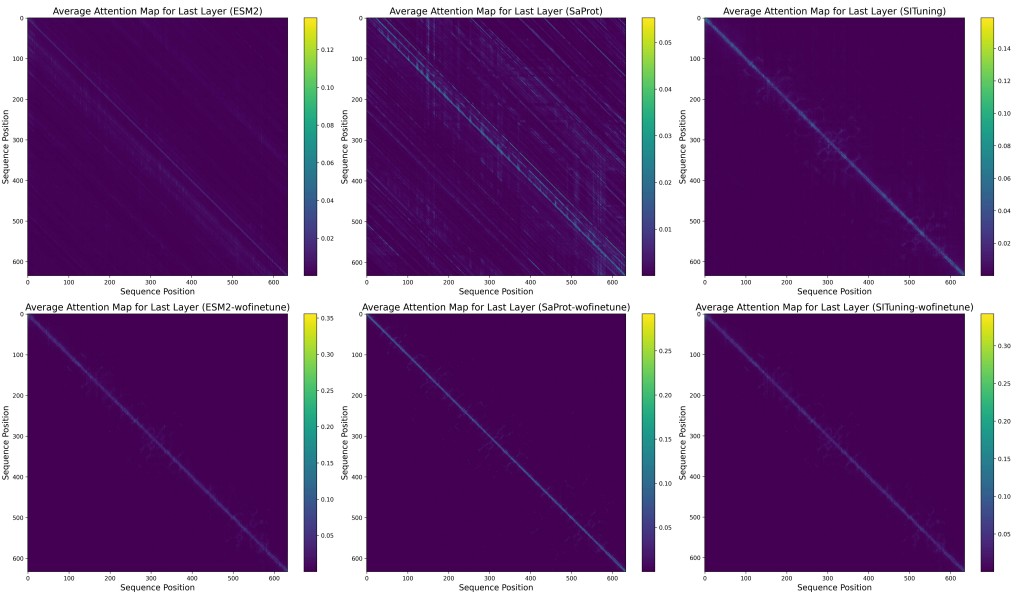

Figure 4: More visualization of attention map of full-parameter tuned ESM-2, SaProt, and our parameter-efficient SI-Tuning.

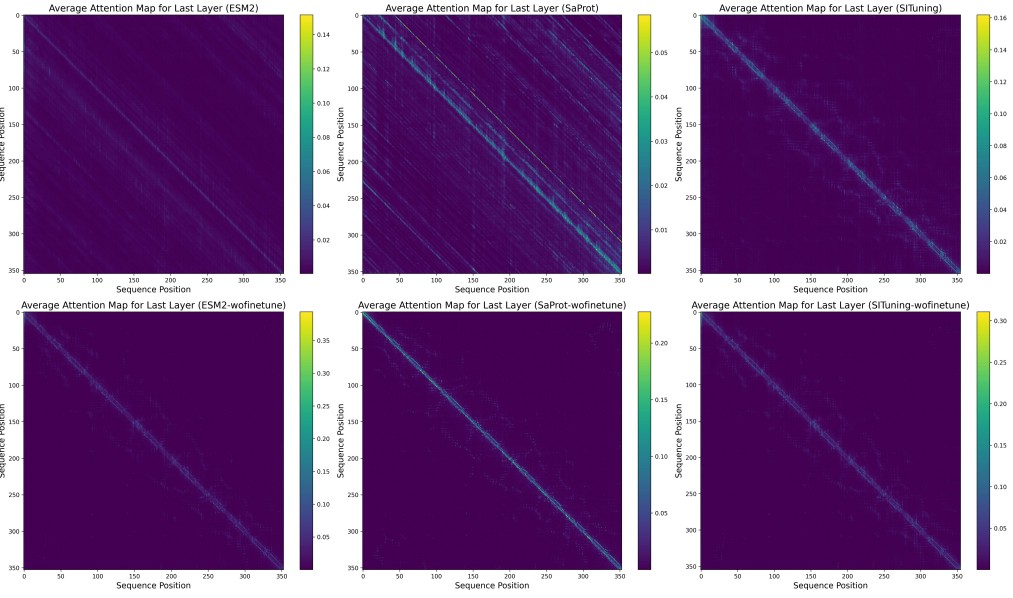

Figure 5: More visualization of attention map of full-parameter tuned ESM-2, SaProt, and our parameter-efficient SI-Tuning.

In the figures above, we visualize the attention maps for additional protein sequences. The visualization shows that full-parameter fine-tuning can partially disrupt the structural perceptual abilities of

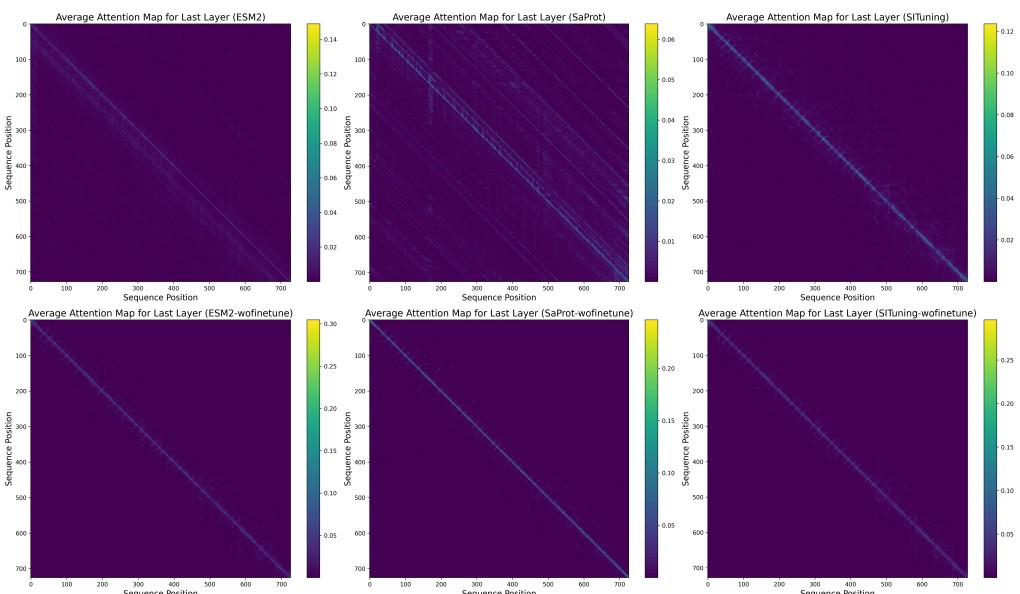

Figure 6: More visualization of attention map of full-parameter tuned ESM-2, SaProt, and our parameter-efficient SI-Tuning.

the pre-trained PLMs. In contrast, our SI-Tuning approach enhances the structural contact patterns in the attention maps by injecting structural information, further improving the model's structural awareness.

## B IMPROVEMENT ACROSS DIFFERENT PLDDT INTERVALS

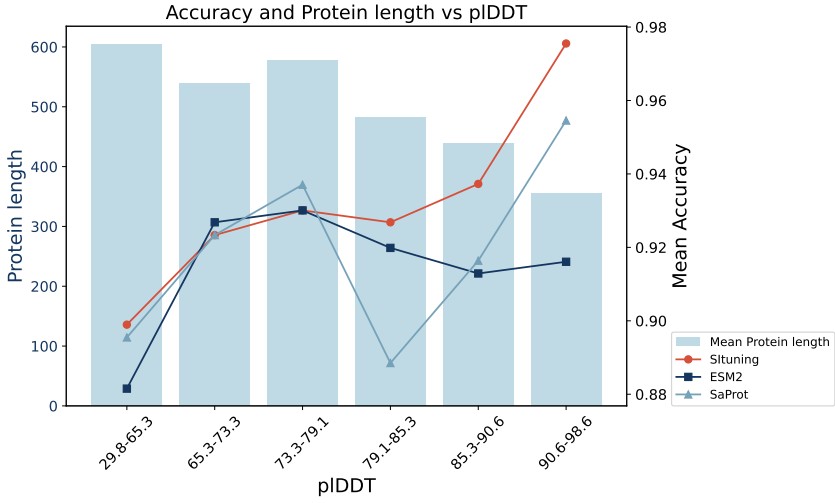

Figure 7: Accuracy of DeepLoc binary classification across different pLDDT intervals.

In Fig, 7, we present the accuracy of different methods across various pLDDT score intervals. The figure demonstrates that, compared to the fully fine-tuned ESM-2, our SI-Tuning shows more significant improvements in intervals with higher pLDDT scores, validating the effectiveness of our structural information injection. Similarly, our SI-Tuning significantly outperforms SaProt in the

higher pLDDT score intervals, highlighting the advantage of incorporating fine-grained structural information over structural vocabulary introduced in SaProt.

## C  RESULTS ON ESM-2 35M

Table 6: Evaluation results of SI-Tuning under ESM-2 35M. *Therm.* and *MIBind.* denote Thermostability and Metal Ion Binding datasets, respectively. *Bin.* and *Sub.* represent Binary and Subcellular, respectively.

| Model | Therm. | MIBind. | EC | DeepLoc | |
|---|---|---|---|---|---|
| | | | | Bin. | Sub. |
| | $\rho$ | ACC | Fmax | ACC | ACC |
| Full tuning | 0.669 | 73.08 | 0.825 | 91.60 | 76.58 |
| SI-Tuning | 0.687 | 74.69 | 0.825 | 92.79 | 79.65 |