# OpenReview forum: "Boost Protein Language Model with Injected Structure Information through Parameter Efficient Fine-tuning"
_ICLR.cc/2025/Conference — ICLR 2025 Conference Withdrawn Submission_

### Official Review · Reviewer_HFbM · 2024-10-19

**Soundness:** 2
**Presentation:** 3
**Contribution:** 2
**Rating:** 3
**Confidence:** 5

**Summary:**

The authors introduced a parameter-efficient fine-tuning framework called SI-Tuning to enhance sequence-based protein language models by incorporating structural information. They conducted multiple experiments to demonstrate the efficacy of their approach. Further, they did serveral ablation studies to investigate the impact of different structural information and hyper-parameters.

**Strengths:**

The authors introduced a straightforward architecture to enhance protein language models with structural information. Through training with a parameter-efficient fine-tuning technique, they improved the model's performance while adding only a small percentage of training parameters.

**Weaknesses:**

1. The novelty is limited. There are many papers about Injecting structural information into protein language models, such as [1-4], etc. Also, the introduction of PEFT technique is not new in this area, such as [5,6]. The combination of these two aspects may not bring new insights for this area.

2. The motivation is insufficient. There are already many open structure-based protein language models, such SaProt, ESM-3 etc, so it seems not that necessary to propose another way to boost sequence-based models. Besides, the pre-training cost of SaProt doesn't affect its downstream applications by fine-tuning.

2. The performance is not outstanding. I don't see some superior results by integrating SI-Tuning, compared to exsiting baselines. Also, the results lack the comparison with other structure-enhanced methods.

references:

*[1] Structure, Surface and Interface Informed Protein Language Model*

*[2] Structure-Informed Protein Language Model*

*[3] Endowing Protein Language Models with Structural Knowledge*

*[4] ProteinAdapter: Adapting Pre-trained Large Protein Models for Efficient Protein Representation Learning*

*[5] Democratizing Protein Language Models with Parameter-Efficient Fine-Tuning*

*[6] Fine-tuning protein language models boosts predictions across diverse tasks*

**Questions:**

1. The authors mentioned in abstract that during training they achieved a reduction of 40.3% GPU memory and 39.8% time consumption, but I didn't see more detailed explanation about this result. To my knowledge, fine-tuning models with common LoRA would't speed up training process greatly. Could you give more information about this result?

---

### Official Review · Reviewer_YiyW · 2024-10-29

**Soundness:** 3
**Presentation:** 3
**Contribution:** 2
**Rating:** 5
**Confidence:** 4

**Summary:**

This paper proposes SI-Tuning, a parameter efficient fine-tuning approach that incorporates structural information into protein language models (PLMs). In the proposed framework, the authors use angular information and pairwise distance to inject the sequence embeddings and the attention map respectively. They also leverage LoRA to efficiently adapt PLM to downstream tasks with structural embeddings. SI-Tuning manages to outperform SaProt and ESM-GearNet in some benchmarks with far fewer trainable parameters and less computational resources.

**Strengths:**

1.	This paper introduces SI-Tuning, an effective and efficient fine-tuning method that injects structural information into PLMs.
2.	The proposed method is clearly written.
3.	The authors conduct comprehensive experiments to demonstrate the effectiveness of this approach.

**Weaknesses:**

1.	The performance of SI-Tuning is not better than SaProt and ESM-GearNet on many datasets. It’s better to discuss the performance-efficiency tradeoff of SI-Tuning in a more detailed manner.
2.	The proposed framework lacks novelty and only uses some common methods to integrate structural information, e.g. more input features and attention bias.

**Questions:**

The need of leveraging PEFT methods (e.g. LoRA) in a 650M ESM-2 is unclear. Does full tuned ESM-2 with structural information perform better? To prove SI-Tuning’s superiority, another comparison between SI-Tuned ESM-2 and LoRA-tuned ESM-2 (or SaProt) is needed.

---

### Official Review · Reviewer_txht · 2024-11-04

**Soundness:** 3
**Presentation:** 3
**Contribution:** 2
**Rating:** 5
**Confidence:** 5

**Summary:**

This paper proposes a novel fine-tuning method called SI-Tuning, designed to inject structural information into Protein Language Models (PLMs) while keeping most of the original model parameters frozen. By incorporating both individual-level (dihedral angles) and pairwise (distance maps) structural data, the model enhances performance in protein-related tasks. The structural data is integrated using embedding and attention map injections, and LoRA is employed for parameter-efficient fine-tuning. Experimental results show that the proposed SI-Tuning approach outperforms other models, like SaProt, while significantly reducing memory and time consumption during training.

**Strengths:**

The paper introduces a well-thought-out and efficient approach for incorporating structural information into Protein Language Models (PLMs). The proposed SI-Tuning method effectively injects structural features, such as dihedral angles and distance maps, into PLMs through embedding and attention map injections, offering a novel way to enhance protein representation without fully retraining the models.

**Weaknesses:**

One issue with this paper is that using LoRA for fine-tuning large models has already become a standard approach, so it is not a novel contribution as presented in the paper.

It would be more beneficial to compare the proposed method with other sequence- and structure-based pretraining approaches, such as those mentioned in the paper (Chen et al., 2023; Guo et al., 2022). Additionally, comparisons with models designed specifically for protein structure tasks, such as ESMFold and AlphaFold3, could provide more meaningful insights.

From the experimental results, the improvements in downstream tasks appear to be relatively modest. More in-depth analysis of why the performance gains are limited would strengthen the evaluation.

Finally, the training code is not available, which limits the reproducibility and further validation of the work.

**Questions:**

Moreover, it is surprising that in most of the downstream tasks, the ESM2 model performs worse than the older ESM1b, which contradicts general expectations and needs further explanation.

---

### Note · Authors · 2024-12-07

I have read and agree with the venue's withdrawal policy on behalf of myself and my co-authors.